# Detecting Equine Gaits Through Rider-Worn Accelerometers

**DOI:** 10.3390/ani15081080

**Published:** 2025-04-08

**Authors:** Jorn Schampheleer, Anniek Eerdekens, Wout Joseph, Luc Martens, Margot Deruyck

**Affiliations:** WAVES, Department of Information Technology, Ghent University-IMEC, 9052 Ghent, Belgium; anniek.eerdekens@ugent.be (A.E.); wout.joseph@ugent.be (W.J.); luc.martens@ugent.be (L.M.); margot.deruyck@ugent.be (M.D.)

**Keywords:** equines, accelerometer sensor, animal activity recognition, machine learning, convolutional LSTM network

## Abstract

Understanding how horses move can improve training practices and support equestrian health, but attaching sensors directly to them may cause discomfort and interfere with natural movement. To overcome this, our study placed sensors on riders instead of horses to classify different horse gaits (halt, walk, trot, and canter). We tested four different sensor placements on the riders—the knee, backbone, chest, and arm—examining five riders and seven horses. Our research also explored how sensor settings, such as data collection speed and analysis interval, affect classification accuracy. After comparing eight different classification models, we found that a specialized neural network model performed best, correctly identifying horse movements with 89.7% accuracy. These findings help show how wearable technology can assist in monitoring horse movement accurately and comfortably, potentially benefiting horse training and welfare.

## 1. Introduction

The market for connected wearable devices has been growing significantly, with 19% growth from 2021 to 2022, reaching a market size of 1.1 billion devices [1]. Human activity recognition (HAR) using wearable devices has been extensively studied, focusing on identifying human activities from sensor data [2]. In contrast, animal activity recognition (AAR) has received less attention. Among the studies centered around AAR, the predominant focus has been on classifying the behavior of cattle [3].

Nonetheless, registering horse movements during training can provide valuable insights into the intensity of the training and enable riders to improve the performance of sport horses [4]. However, most AAR studies rely on multiple sensors attached to the animals. For example, in a previous study [5], the movements of horses were monitored using five sensors attached to each leg, including accelerometers and a GPS sensor. Another study [6] used eight sensors, each equipped with an accelerometer, a gyroscope, and a compass, on the poll, withers, sacrum, and sternum and each limb. The sampling rate in the latter study ranged from 100 Hz to 200 Hz, resulting in high battery consumption. A study conducted at Ghent University [7] tackled this problem by attaching an accelerometer to the horse’s leg, achieving 100% accuracy in classifying jumping training and 96.29% accuracy in classifying dressage training. The solution involved using two accelerometers attached to the horse’s front legs, with no specialized equipment required, as the sensors could be fastened with Velcro to the already attached tendon boots.

However, the willingness of many riders to attach wearables to the horse is limited due to concerns about discomfort, behavioral disruption, and entanglement risks. In a study conducted at the University of Kentucky [8], these issues were addressed by utilizing the accelerometer of a smartwatch worn by the rider, eliminating the need for horse-worn equipment. This study achieved 92% accuracy in distinguishing between walk, trot, and canter gaits. However, due to the limitations of the smartwatch, only the wrist could be used to evaluate sensor data, and no other positions could be considered.

The aim of the present study is to address the aforementioned issue by attaching accelerometers (Axivity AX6) to the rider instead of the horse. This approach is expected to increase riders’ willingness to use the sensors since there is less risk of injury or altered behavior, as associated with sensor attachment to the horse. To this end, an experimental setup is proposed to find the optimal sampling rate, time window, sensor location on the body, and classification model. These parameters are evaluated since they immediately influence the accuracy of the classification.

The novelty of this paper lies in the evaluation of eight different models across thirteen window sizes, four different sensor locations on the rider, and four frequencies to classify four horse movements (halt, walk, trot, and canter) during training.

## 2. Related Work

The use of accelerometer data for recognizing human and animal activities has gained significant attention in recent years due to its broad applications, such as in wildlife monitoring, human health tracking, and behavioral analysis. This section provides an overview of the studies conducted in the field of activity recognition since 2020. These studies were selected based on their use of accelerometer data for activity recognition. Table 1 summarizes the papers and related work. A number of these studies concentrated on recognizing human activities using accelerometer data. For example, Alessandrini et al. [9] and Agarwal and Alam [10] achieved an accuracy of 95.54% and 95.78%, respectively, in human activity recognition using Long Short-Term Memory (LSTM) networks. Mekruksavanich and Jitpattanakul [11] combined Convolutional Neural Networks (CNNs) with LSTM, resulting in an accuracy of 95.86%. This approach highlights the advantages of combining both spatial and temporal features. Additionally, studies such as those by Fridriksdottir and Bonomi [12] further demonstrated the effectiveness of hybrid models, specifically the fusion of CNNs and LSTM, achieving an accuracy of 94.52%.

The application of accelerometer data for recognizing animal activities has also been explored. Conners et al. [13] employed a Hidden Markov Model (HMM) to classify albatross activities, with an accuracy of 91.6%. This study showcased the effectiveness of using probabilistic models to capture complex behavior patterns. Hussain et al. [14] effectively used Long Short-Term Memory (LSTM) networks to classify dog activities, with an accuracy of 94.25%. Similarly, Kasnesis et al. [15] employed Convolutional Neural Networks (CNNs) to identify dog activities, with an accuracy of 91.26%.

In the context of livestock, Schmeling et al. [16] utilized Random Forest models for cow activity recognition, achieving an accuracy of 87.3%. Bloch et al. [17] further elevated accuracy by applying CNN-based cow activity recognition, attaining an accuracy of 93.3%. Multilayer perceptron (MLP) was also found useful for recognizing animal activities in studies such as those by Voorend [18] and Arablouei et al. [19]. They reached an accuracy of 86% and 87.62%, respectively. An overview of these studies is presented in Table 1.

Regarding studies that performed horse activity classification, Eerdekens et al. [20] combined data from both ridden horses and longed horses. Birant and Tepe [21] and Voorend [18] combined data from both ridden horses and horses that roamed freely in a pasture. Casella et al. [8] collected data from ridden horses.

**Table 1 animals-15-01080-t001:** Comparison of machine learning models in the literature for activity recognition using accelerometers.

Author	Data Origin	Year	Best Algorithm	Number of Subjects	Number of Activities	Results
Conners et al. [13]	albatross	2021	HMM	29	3	Accuracy: 91.6%
Hussain et al. [14]	dog	2022	LSTM	10	10	Accuracy: 94.25%
Kasnesis et al. [15]	dog	2022	CNN	7	4	Accuracy: 91.26% F1: 91.21%
Arablouei et al. [19]	cow	2023	MLP	8	6	Accuracy: 87.62%
Bloch et al. [17]	cow	2023	CNN	21	3	Accuracy: 93.3% ± 2.0% F1: 93.9 ± 1.9%
Schmeling et al. [16]	cow	2021	Random Forest	N.A.	2	Accuracy: 87.3%
Hysenllari et al. [22]	human	2022	CNN	35	6	Accuracy: 99.28%
Alessandrini et al. [9]	human	2021	LSTM	7	3	Accuracy: 95.54%
Mekruk-savanich, Jitpattanakul [11]	human	2021	CNN + LSTM	14	12	Accuracy: 95.86%
Agarwal, Alam [10]	human	2020	LSTM	29	6	Accuracy: 95.78% F1: 95.73%
Fridriksdottir, Bonomi [12]	human	2020	CNN + LSTM	20	6	Accuracy: 94.52% F1: 94.64%
Birant, Tepe [21]	horse	2022	Extra Forest	18	5	Accuracy: 94.62%
Eerdekens et al. [20]	horse	2021	Hybrid-CNN	6	7	Accuracy: 99.59%
Voorend [18]	horse	2021	MLP	18	18	Accuracy: 86%
Casella et al. [8]	horse	2020	k-NN	2	3	Accuracy: 92%
Kleanthous et al. [23]	sheep	2022	CNN	9	3	Accuracy: 98.55%

When investigating the choice of sampling frequency across the studies, a wide variation was found. For instance, Kleanthous et al. [24] compiled data on sheep behavior classification studies and found sampling frequencies ranging from 1 Hz to 200 Hz in the literature. The optimal sampling frequency depends on factors like the activities being classified, the animal species, the sensor location, and the chosen model. For example, Walton et al. discovered that 32 Hz yielded the highest accuracy, while Kleanthous et al.’s best performing study used a sampling rate of 12.5 Hz. For horse studies, Eerdekens et al. [20] identified 10 Hz as the optimal sampling frequency for sensors attached to the horse’s front legs.

Similarly, the selection of interval width also varies. Decandia et al. [25] found 30 s intervals optimal for sheep behavior classification, whereas Alvarenga et al. [26] demonstrated that 5 s intervals were the most effective. This emphasizes the challenge of comparing interval widths and sampling frequencies across different studies, particularly when considering different models, animal classes, sensor placements, and activity types.

## 3. Materials and Methods

### 3.1. Data Collection

Data were collected from October 2022 to January 2023 at a local stable in Neeroeteren, Belgium. In total, five riders rode seven horses during 11 different activities (halt; collected, medium, and normal walk; collected, medium, and normal trot; and collected, medium, and normal canter) [5,27,28]. The riders had all been riding since a young age and had at least 7 years of experience. However, none of them performed competitively. All riders identified as female, had a height between 1.64 m and 1.75 m, and weighed between 60 and 70 kg. The riders were assigned to the horses that they typically rode, and they were encouraged to ride as they normally would. They performed rising trot and sitting canter. Two recording sessions were planned for each rider. For this reason, four Axivity AX6 accelerometers (Axivity Ltd., Newcastle upon Tyne, UK) were used to collect data simultaneously. Using medical tape, the sensors were attached to the rider on the knee, the backbone, the chest, and the arm. The sensors were worn under the clothes, which meant that, even if they became loose, they would pose no risk to the horse by falling. Another advantage of this was that the clothes already secured the sensor in place, and the medical tape was only necessary to reduce noise in the signal and keep the sensor still. This meant that the medical tape did not have to be secured tightly in a way that it could impact the comfort of wearing the sensor. Figure 1 shows the sensor placement and orientation.

The sensors were oriented in such a way that the x-axis always pointed towards the ground, and the z-axis always pointed away from the rider. This is indicated in Figure 1. The sensors were configured to sample data at 50 Hz because it was proven by Pfau et al. [29] that this is the lowest frequency that contains all necessary signal information for horse gait recognition to detect gait asymmetry, which requires millimeter precision. For this reason, 50 Hz served as an upper limit for the sampling rate. The riders rode for about 30 min to 1 h at a time while being filmed with an iPhone 13 Pro (Apple Inc., Cupertino, CA, USA) from the corner of the arena. In total, 8 h and 29 min of data were collected. An overview of the horses that participated in this study with the collected samples per gait per horse is presented in Table 2. Of the seven horses that participated in this study, five were different breeds: Lusitano, a Portuguese breed; PRE, a Spanish breed; BWP, a Belgian breed; and English Pony, an English breed. The decision to include different breeds served to capture the diversity in equine locomotion patterns and behavior. This decision was motivated by the recognition that different horse breeds exhibit distinct gait mechanics, which can significantly impact inertial signals, as demonstrated by Rhodin et al. [30]. Since the riders were encouraged to ride naturally, the rider of the horse Tango never felt the need to halt. For this reason, no halt samples were collected for Tango. In total, 1,324,324 usable samples were collected, resulting in 7 h and 21 min of data.

### 3.2. Data Preprocessing

After the data collection, the video data and accelerometer data were downloaded in the file formats MP4 and CSV, respectively. The sensors were equipped with a real-time clock module (RTC) that was synced with the timestamp on the videos with millisecond precision. In order to be usable for the machine learning algorithm, the accelerometer data needed to be annotated using the video data. For this purpose, ELAN [31] was used. The video file was loaded into ELAN, and, for each time interval, labels were entered. The labels were based on verbal cues from the riders themselves in the video recording. The resulting dataset served as the ground truth for the machine learning algorithm.

### 3.3. Machine Learning Model Selection

Based on the literature study results in Table 1, a list of useful models could be compiled. While simple data processing techniques have been shown to achieve high accuracy, such as in Pfau and Guire [32], it has been shown that, in general, deep learning models are able to achieve higher accuracies than traditional machine learning techniques [33]. For this reason, the choice was made to focus only on deep learning models. According to the literature study, the following deep learning models have been proven to perform well in activity classification: Convolutional Neural Networks (CNNs), Long Short-Term Memory (LSTM), Hybrid-CNN, and multilayer perceptron (MLP). More specifically, the following eight models were implemented in this study to perform classification: an MLP, a CNN, a dilated CNN, a CNN with Spatial Dropout instead of regular dropout, a small LSTM, a large LSTM, a network based on the Wavenet [34] architecture, and a convolutional LSTM. These models were implemented as follows:MLP: An MLP with two hidden layers, consisting of 128 and 64 neurons, and an output layer of 4 neurons. This model has 26,702 trainable parameters.CNN: A CNN with two 1D convolutional layers with eight filters, followed by a max pooling layer and then another convolutional layer with eight filters, and ending with a dropout layer and a dense layer, with 28 neurons in the hidden layer and 4 in the output layer. The entire network is defined by 3162 trainable parameters.Dilated CNNs: A CNN with two 1D convolutional layers with seven filters, followed by a max pooling layer and then another convolutional layer with seven filters, and ending with a dropout layer and a dense layer, with 42 hidden neurons and 4 output neurons. The second convolutional layer uses dilations to broaden its overview. This model has 7084 trainable parameters.Spatial Dropout CNN: A CNN with four 1D convolutional layers of 120 filters, a kernel size of 15, and dilations, each followed by a Spatial Dropout layer with a probability of 5%, ending with an LSTM with 256 cells and a dense layer with 512 hidden neurons. This model consists of 1,178,694 trainable parameters.Small LSTM: An LSTM network with two LSTM layers, each with four LSTM cells, followed by a dropout layer and a dense layer with 64 hidden neurons and 4 output neurons. This model has 342 trainable parameters.Large LSTM: An LSTM network with two LSTM layers, each with 32 LSTM cells, followed by a dropout layer. This network consists of 13,390 trainable parameters.Wavenet: A network based on the Wavenet [34] architecture. The model is based on dilated convolutional layers. These convolutional layers are connected with skip connections, residual connections, and gates. The gates allow the model to determine which information to carry over to other layers, while the skip connections and residual connections prevent the gradient from vanishing or exploding during training. These connections also allow Wavenet to go deeper than classic neural networks, resulting in very good performance in most use cases. This model consists of 509,758 trainable parameters.Convolutional LSTM: An LSTM network fused with convolutional operations. One convolutional LSTM layer with eight filters is followed by a dropout layer and ends with a dense layer with 64 hidden neurons and 4 output neurons. The model is defined by 32,014 trainable parameters.

### 3.4. Test Setup

A test setup was needed to compare the selected models and find the ideal window size, sampling rate, and sensor location. A grid search allowed for this experiment to be performed by comparing all combinations of hyperparameters and models. The test setup consisted of four parameters: eight deep learning models, four sensor locations, thirteen window sizes, and four frequencies. In the test setup, the window sizes considered were 0.6, 1, 1.2, 1.6, 2, 2.2, 2.6, 3, 3.2, 3.6, 4, 5, and 6 s. For the resampled frequency, the considered frequencies were 5 Hz, 10 Hz, 25 Hz, and 50 Hz. These were chosen because they are all denominators of 50, enabling the resampling to work without interpolation. After conducting all experiments, the data of each experiment were scaled using RobustScaler, which removes the median and scales the data according to the quantile range. Every model in the test setup was trained for 15 epochs on a machine with 16 GB RAM, an Intel (Intel Ireland Ltd., County Kildare, Ireland) i7-5820K hexacore processor overclocked to 4.20 GHz, and an NVIDIA (NVIDIA CORPORATION, Wilmington, DE, USA) GeForce GTX 970 with 4 GB VRAM.

### 3.5. Window Size Selection

The deep learning models were trained using pure data, which meant that they only saw intervals of data corresponding to specific horse activities. This approach improved the model’s learning reliability. However, it had a limitation: when using a ten-second window size, the training data only included activities that lasted for at least ten seconds continuously. Some activities were exclusively ridden on the long side of the arena, while the short side was used for variation. As a result, these particular activities were not included in the training data when using a ten-second window size. Although the results are representative and significant, indicating that the model may perform better with fewer activities, the goal in this case was to retain data from all activities. Therefore, the median time of each activity in the training data was determined to be 6.3 s. Hence, to maintain sufficient data in this dataset, a window size smaller than 6.3 s was required. For this reason, window sizes between 0.6 and 6 s were chosen.

## 4. Results

As discussed in the previous section, four parameters were analyzed using grid search. A comparison of these results led to an optimal configuration. When deciding the optimal value of a parameter, either the average accuracy of the parameter is used or the maximum achieved accuracy is used, depending on the type of conclusion being drawn. More explicitly, when deciding on a general parameter, the average accuracy is used. For example, when choosing the optimal classification model, the average accuracy is used since it is important to choose a model that fits the data well in general and not in one case with a high maximum accuracy. This guarantees the affinity of the model with the data. In cases when a parameter depends on another parameter, the maximum accuracy is used, for example, when fixing the classification model and finding the optimal sampling frequency for this specific model. This is because, in this case, it is not important to find a general well-performing value but rather the best performing value for the fixed parameter. For general parameters, error bars can be plotted because they provide meaningful comparisons. However, for parameters that depend on other parameters, plotting error bars is not feasible, as they would be influenced by the variability of the dependent parameters, leading to inaccurate representations. For instance, when selecting the optimal frequency, the error bars for that frequency would reflect the performance across all time windows. However, it is evident that the optimal frequency is defined by a single time window where it performs best.

Figure 2 illustrates this process. First, the collected samples were split into a training set and a validation set. The validation set comprised all data from the horse Tango, and the training set contained all other data. Tango was chosen since his rider had the most experience. Tango also likes riding, which caused him to never protest and always make clear changes in gaits. This combination made Tango a good choice for the validation. Second, a permutation was calculated for all possibilities of models, sensors, window sizes, and sampling frequencies. This resulted in 1664 configurations (8 models, 13 window sizes, 4 sampling frequencies, and 4 sensor locations) being evaluated using the validation set. These data were then used to propose the optimal configuration. For example, to determine which model was able to learn the most information regardless of how the data were structured (no matter the location, window size, or frequency), the average was taken from these accuracies for each model and then compared.

### 4.1. Optimal Model

To identify the best model, the average performance of each model over all window sizes and frequencies was compared. Figure 3 shows this comparison.

For clarity, these models are divided into two groups based on their performance. The first group contains the underperforming models. These models fail to reach an average score higher than 79%. Based on the information in the graph, this group consists of the MLP network (“ONLY DENSE”, 75.56%), the small LSTM network (“PURE LSTM”, 67.86%), the large LSTM network (“LSTM DENSE”, 60.07%), the spatial dropout network (“SPATIAL DROPOUT”, 78.93%), and the Wave network (“WAVENET”, 71.99%). The second group consists of three models, whose average accuracy is better than 79%: the convolutional network (“CONV1D + DENSE”, 79.67%), the dilated convolutional network (“DILATED CNNS”, 79.85%), and the LSTM-convolutional network (“CONVLSTM2D”, 80.37%). These are the good performing networks. For brevity and relevance, only the good performing networks are considered in the rest of this paper.

### 4.2. Optimal Window Size

Figure 4 shows the maximum accuracy achieved by the well-performing models per interval.

In this case, the LSTM-convolutional model achieves the highest accuracy at a window size of four seconds (90.73%), while the convolutional and dilated convolutional networks achieve their highest accuracy at a window size of five seconds (89.57% and 89.36%, respectively). For the LSTM-convolutional model, the maximum accuracy at an interval width of four seconds is 16.48% higher than the maximum accuracy at 0.2 s. For the considered interval widths, the average increase in accuracy per extra second in the interval is 2.54%. In conclusion, for the LSTM-convolutional model, the optimal window size is four seconds, while for the convolutional and dilated convolutional models, the optimal window size is five seconds. Since the LSTM-convolutional model is found to outperform the convolutional and dilated convolutional models in Section 4.1, we suggest the use of the LSTM-convolutional model with a window size of four seconds for the best results.

### 4.3. Optimal Sampling Rate

For the sampling rate, determining a single optimal frequency is not feasible, as it varies across models. However, the best frequency for each model can be analyzed. Figure 5 shows the maximum accuracy per model per frequency for intervals with a width less than or equal to six seconds.

The LSTM-convolutional network achieves its highest accuracy at a frequency of 25 Hz (90.73%), while both convolutional models perform best at 10 Hz (89.57% for the convolutional model and 89.36% for the dilated convolutional model). For the LSTM-convolutional network, going from 10 Hz to 25 Hz results in a 2.63% increase in maximum accuracy, while for the convolutional and dilated convolutional models, going from 10 Hz to 25 Hz results in a 1.16% and 0.57% decrease in maximum accuracy, respectively. The drop in accuracy at the 50 Hz sampling frequency for the LSTM-convolutional and dilated convolutional models shows that these models start overfitting when too much data are available. In conclusion, we suggest the use of the LSTM-convolutional model with a window size of four seconds and a sampling frequency of 25 Hz for the best results.

### 4.4. Optimal Sensor Location

Finally, the optimal sensor location is examined. Considering the average accuracy across frequencies, models, and intervals, it is suggested that placing the sensor on the knee yields the highest accuracies.

Zooming in on the graph allows one to focus on the best models and their corresponding frequencies. Figure 6 shows the accuracy of the top-performing networks, averaged over frequencies of 10 Hz and 25 Hz.

Using an independent samples test based on the Welch *t*-test with α=0.05, it is found that the knee sensor achieves higher accuracy than the chest (p<0.001) and backbone sensors (p=0.01), but it can not be stated with enough confidence that the knee sensor achieves higher accuracy than the arm sensor. The good performance of the knee sensor, as shown in Figure 7 (76.23%), can be attributed to the concept of the center of mass in equine biomechanics [35,36,37]. The center of mass summarizes the horse’s movements, reducing the influence of noise. Based on this concept, the sensor closest to the horse’s center of mass, specifically the knee sensor, exhibits better accuracy. The reason behind the superior performance of the arm sensor (75.77%) compared to the chest (71.40%) and backbone (73.37%) sensors is not conclusively determined. One plausible explanation is the riding convention of maintaining stable hands relative to the horse to avoid unintentional pulling of the reins [38]. This practice requires keeping the shoulders and elbows relaxed, acting as a motion stabilizer and enabling the arm sensor to better track the horse’s movements. However, the reliability of the arm sensor data may depend on the rider’s experience, as novice riders often struggle to keep their hands steady. The riders in this study were not novice, which explains the good accuracy of the arm sensor results. Notably, the arm sensors were placed on the upper arm, which is a more stable location than the lower arm and wrist. This can also be seen in a study by Eisersiö, who found a correlation between hand and head movements of the horse during trot [39]. Therefore, we suggest the use of the knee for the location of the sensor, as it would not be affected by the rider’s experience.

### 4.5. Optimal Setup

Finally, after determining the LSTM-convolutional model as the optimal model, 25 Hz as its optimal sampling frequency, four seconds as the optimal window size, and the knee as the optimal sensor location, the results of this optimal setup can be verified. The results are verified using Leave One Subject Out Cross-Validation (LOSOCV), where the data are automatically split into training and validation sets based on the horse. Each horse is kept separate in the validation set. To ensure the purity of the validation data, all sessions involving a particular horse are used exclusively for validation and removed from the training data. Due to the limited data available for Dizzie and Ruby (4.39% and 6.34% of the total collected data, respectively), they are not included in the cross-validation process. The remaining five horses (Tango, Elmo, Galand, Jaleo, and Queenie; see Table 2) are alternately used to validate the model’s performance.

### 4.6. Optimal Setup Verification Results

In the verification setup, the LSTM-convolutional model is verified using LOSOCV with a four-second window size and a sampling frequency of 25 Hz. Four gait classes are considered: halt, walk, trot, and canter. Table 3 presents the results of this experiment as accuracies from the cross-validation. On average, this model achieves an accuracy of 89.72% when including Galand in the LOSOCV process and 98.31% when excluding Galand from being used in the validation. Moreover, this model achieves an average F1-score of 86.18%.

### 4.7. Overall Performance

When using the optimal configuration of the convolutional-LSTM, with the data sampled at 25 Hz with a window size of four seconds by a sensor on the knee of the rider, as well as employing four superclasses, the model achieves an average accuracy of 89.72% and an F1-score of 86.18%. However, when including Galand’s data in the training dataset, the model reaches an average accuracy of 98.31%, complemented by an F1-score of 96.21%.

### 4.8. Summary

Throughout this study, accelerometers attached to riders were used to automatically identify horse activities using machine learning models. We defined four variables to identify an optimal experimental setup and verified the results using cross-validation. Table 4 presents a summary of this optimal setup for four classes.

## 5. Discussion

### 5.1. Models

The ConvLSTM2D model emerged as the optimal outcome from the grid search (Table 4). Heroy et al. [40] previously demonstrated the high efficacy of this model in human activity recognition. In their work, the LSTM-convolutional model was compared against a convolutional model, an LSTM model, a bidirectional LSTM model, and an MLP model. The findings highlighted the LSTM-convolutional model as the superior performer, with the convolutional model in the second-best position—a trend analogous to the outcomes observed in the present study. Likewise, a study conducted by Vesa et al. [41] similarly found that the LSTM-convolutional network was the optimal choice for human activity recognition after a comparative evaluation against an LSTM model and a hybrid-CNN. As such, the current investigation concludes that the LSTM-convolutional network outperforms not only other established networks in human activity recognition but also in the context of recognizing equine activities using sensor data collected from the rider. To the best of the authors’ knowledge, there exist no publications utilizing this particular model for animal activity recognition. It is noteworthy that the model seems to generalize well despite the diverse dataset, consistently achieving the same accuracies. For instance, Jaleo is only 155 cm tall, while Galand is 173 cm, and Elmo is a Portuguese horse, whereas Galand is a Belgian horse. Despite this, using Galand to train the validation of the generalization towards these different horses proves positive.

### 5.2. Best Window Size

The grid search procedure revealed that, for the LSTM-convolutional model, a frequency of 25 Hz yielded the best accuracy. In a study conducted by Eerdekens et al. [20], involving the recognition of equine activities through sensors attached to the front legs, a frequency of 10 Hz was identified as the optimal setting for both convolutional and hybrid-convolutional networks. Notably, the present study also found that 10 Hz was the most advantageous frequency for the convolutional network, which was also used in the study by Eerdekens et al. [20].

### 5.3. Sampling Rate

The grid search analysis indicated that an interval width of four seconds yielded the highest accuracy for the LSTM-convolutional model. In a study by Casella et al. [8], where equine activity was classified based on sensor data from a smartwatch, a ten-second interval width was determined to be optimal. However, within the scope of this study, the dataset’s characteristics rendered interval widths exceeding 6.3 s impractical due to substantial data sample loss. Other studies, such as those by Eerdekens et al. [20], Kamminga et al. [42], and Kleanthous et al. [23], adopted a two-second interval width. Kleanthous et al. justified this choice under the premise that animals frequently alter their activities, and excessively large intervals might lead to compromised classification accuracy. It is worth mentioning that Kleanthous et al.’s study [23] pertained to sheep activities in a pasture, where movements are less predictable. Additionally, this logic can also be extended to horseback riding, where during a dressage session, the same gait is exercised for longer periods of time, whereas when horses are in a pasture, their movement is less predictable. In a separate study, Kleanthous et al. [43] conducted a comparison of the interval widths employed across the literature for classifying sheep activities. This analysis revealed substantial variations, ranging from 5 s (Alvarenga et al. [26]) to 30 s (Decandia et al. [25]). This diversity clearly shows the link between interval width selection, classification methodology, model architecture, and dataset characteristics.

### 5.4. Sensor Location

In the literature study on activity recognition, the utilization of rider-mounted sensors was observed in merely one study by Casella et al. [8], who employed a wrist-worn smartwatch. Similarly, in a paper published before 2020, Maga and Björnsdotter [44] employed a smartphone attached to the back. However, both of these studies encountered limitations concerning sensor placement due to considerations of size or shape. Because of this, the optimal sensor placement on the rider’s body could not be definitively determined. This study aimed to address this gap by comparing sensor locations, including on the knee, arm, chest, and backbone, to determine the most effective sensor location for data collection. Through this investigation, the knee was determined to be the optimal location.

### 5.5. Overall Performance

The average accuracy when including Galand in the validation set and using four classes is better than that of 19% of the studies included in the literature study in Section 2. Galand’s influence on the results of the validation set was observed during the cross-validation. Ultimately, the average accuracy when excluding Galand from the validation set surpasses the results of 81% of the studies included in the literature study. The reason why Galand has a significant impact on the model’s accuracy is that Galand constitutes the largest portion of the dataset (23.67% of all data in the dataset come from Galand). Additionally, Galand provides a substantial amount of information on specific activities, for example, on halt (accounting for 55.56% of all halt data). Therefore, when these data are removed from the training set, the model’s ability to learn is significantly hindered.

## 6. Conclusions

The optimal setup of the parameters evaluated in this study was identified for classifying horse activities with sensors placed on the rider, which can be useful for both horse health monitoring and improving performance in equestrian sports. A total of eight models were compared after training on a dataset of seven horses and four possible activities. This study also investigated the optimal sampling frequency, window size, and sensor location. The results show that the LSTM-convolutional model performed best, achieving an average accuracy of 89.72% and an F1-score of 86.18%, which was validated through LOSO cross-validation with a four-second window size and a sampling frequency of 25 Hz. However, this average was significantly lowered because the horse with the most data in the dataset was used for validation and removed from the training set. When this horse was included in the training set and not in the validation set, the model achieved an accuracy of 98.31% and an F1-score of 96.21%. The best sensor location was proven to be the knee of the rider.

Future work will consist of collecting more data and adding more horses to the dataset. This enrichment of the dataset will balance it and allow the model to better discern the differences between the activities. It will also balance the impact of excluding Galand from the training data. Additionally, the data will be further preprocessed, for example, by adding filters to eliminate noise and by integrating manually computed features to aid in classification. This could further improve the accuracy of the model. Furthermore, data from different sensor locations can be combined. When studying this combination, it may be possible to make the model more robust; however, more work is required to support this. The findings of this paper are expected to aid in the development of wearable horse activity recognition applications and serve as the basis for a framework for horse activity recognition.

## Figures and Tables

**Figure 1 animals-15-01080-f001:**
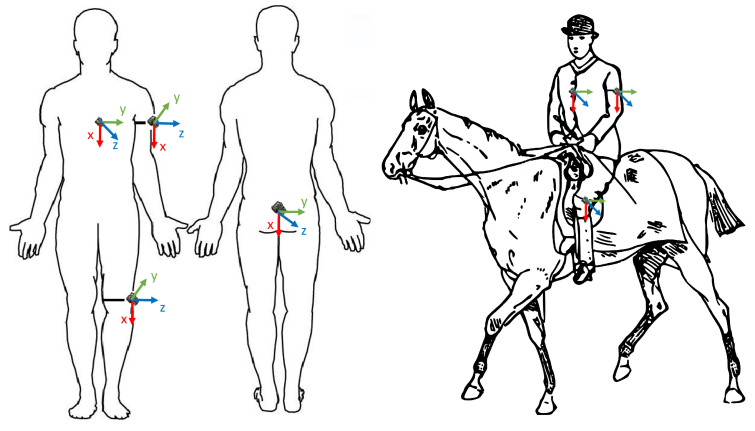
Sensor placement and orientation on riders both on and off the horse.

**Figure 2 animals-15-01080-f002:**
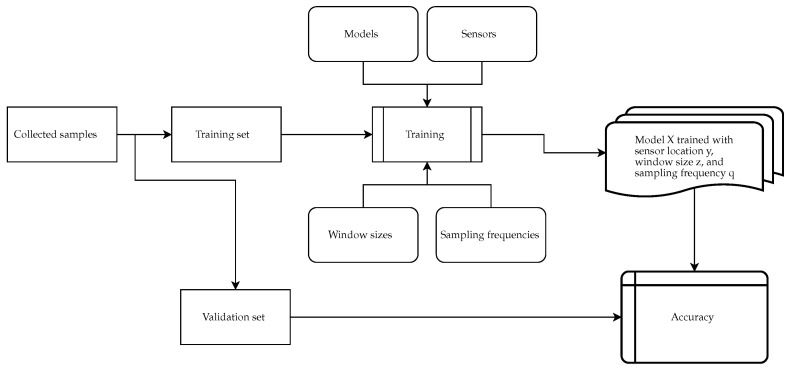
Illustration of the evaluation process.

**Figure 3 animals-15-01080-f003:**
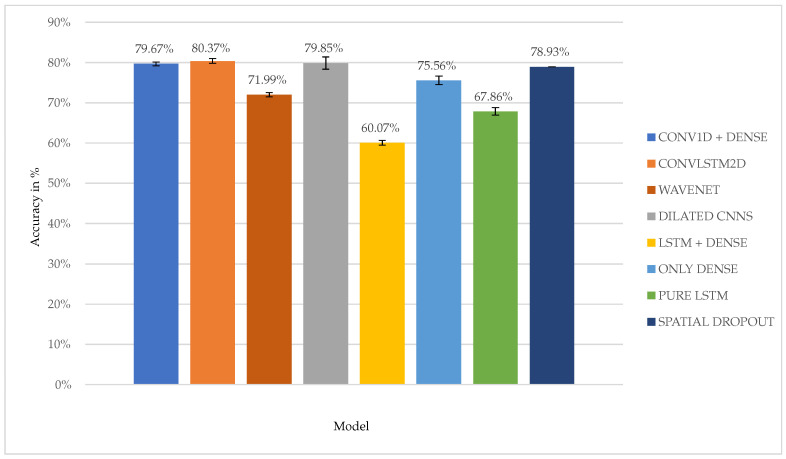
Illustration of the average performance for each model (details of the models in the legend are explained in Section 3.3).

**Figure 4 animals-15-01080-f004:**
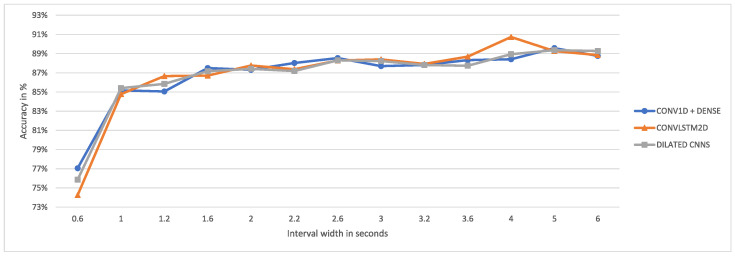
Illustration of the maximum accuracy achieved by the well-performing networks per interval.

**Figure 5 animals-15-01080-f005:**
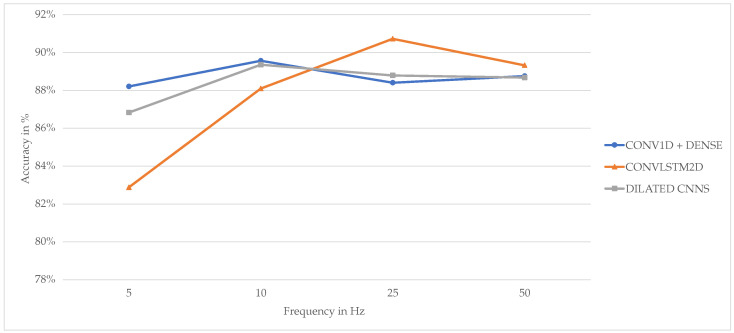
Maximum performance of models for each frequency.

**Figure 6 animals-15-01080-f006:**
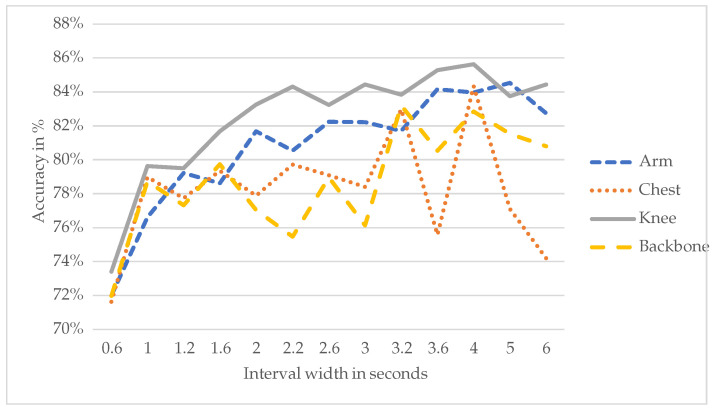
Average performance of the good performing networks averaged over 10 Hz and 25 Hz by sensor location.

**Figure 7 animals-15-01080-f007:**
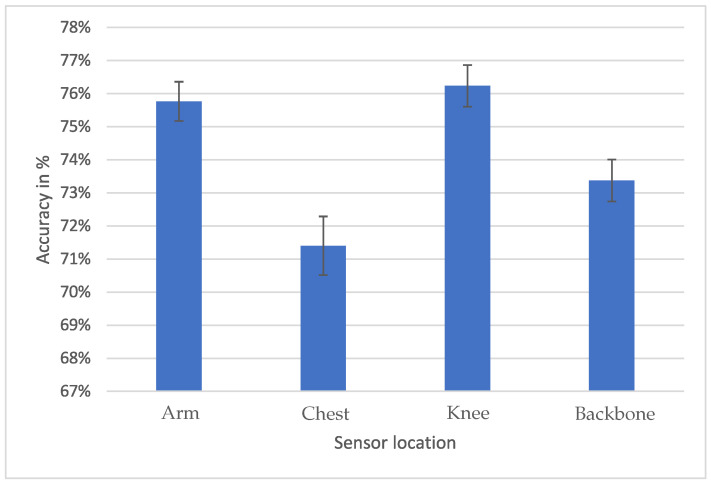
Average performance of the different sensor locations on the rider’s body, namely, the arm, the chest, the knee, and the backbone.

**Table 2 animals-15-01080-t002:** Participating horses, along with their breed and height at withers and the number of samples per collected gait.

Name	Breed	Height at Withers (cm)	Halt (Samples)	Walk (Samples)	Trot (Samples)	Canter Samples)
Dizzie	French Trotter	163 cm	739	37,439	18,072	1830
Elmo	Lusitano	163 cm	8025	103,978	156,844	42,486
Galand	BWP	173 cm	25,025	136,684	99,952	51,765
Jaleo	PRE	155 cm	5507	99,672	59,255	15,374
Queenie	BWP	165 cm	4311	85,846	116,417	38,204
Ruby	English Pony	135 cm	1438	41,675	36,066	5925
Tango	BWP	163 cm	0	69,975	37,727	24,193
			45,045	575,269	524,333	179,777

**Table 3 animals-15-01080-t003:** Results of LOSOCV for the LSTM-convolutional model with knee-mounted sensor, using a four-second interval and a sampling frequency of 25 Hz and considering four classes.

Horse in Validation Set	Accuracy	F1
Tango	98.96%	98.41%
Elmo	96.47%	92.53%
Galand	55.37%	46.45%
Jaleo	99.17%	98.37%
Queenie	98.63%	95.52%

**Table 4 animals-15-01080-t004:** Parameters, accuracy, and F1-score of the optimal experimental setup.

Parameter	Value
Model	ConvLSTM2D
Input vector size	100 (4 s at 25 Hz)
Output vector size	4
Number of filters	8
Filter size	(1, 1)
Stride	(1, 1)
Dropout probability	0.5
Maximum epochs	20
Optimizer	Adam
Accuracy	89.72%
F1-score	86.18%

## Data Availability

The raw data supporting the conclusions of this article will be made available by the authors on request.

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
