# Peer review of "Detecting Equine Gaits Through Rider-Worn Accelerometers"

_animals, 2025, doi:10.3390/ani15081080_

Round 1

Reviewer 1 Report

Comments and Suggestions for Authors

Excellent study. Well written and presented. May aid in future use of AI and rider wearable accelerometers to help in horse training. 

I did have two issues that should be addressed and then a few more comments in attached PDF of paper.

Main Issues to Address

1. I'm not sure about the title. The title starts with Equine IoT but then the abbreviation IoT is not mentioned in paper at all and there are two possible meanings for it that I can see (probably Internet of Things is meant by authors). But if this is meaning it needs to be presented in paper also, and the article should discuss how rider-worn accelerometers could be attached to such a network. I would suggest replacing IoT with another term or phrase as the study mainly focuses on using accelerometer data and setting up IoT would be another step. Alternatively the abbreviation IoT could be mentioned in article and its meaning noted, but I think the former course is probably better. See attached file here also.

2. There is need of some references to studies of horse gaits and their classification. I have noted a couple of sources that could be referenced. See attached file.

Then there are the minor comments in attached file.

Author Response

Thank you very much for taking the time to review this manuscript. We would like to thank you for your insightful, constructive and professional comments.

Please find the detailed responses below and the corrections highlighted in the re-submitted files.

Comment 1: What do you mean by IoT (Internet of Things) (Intensity of Training). I would spell out what this abbreviation means in introduction of article at least and perhaps in title. And I would say 'through rider-worn accelerometers'.

Response 1: We have changed the title to better reflect the content of this work to:
“Detecting equine gaits through rider-worn accelerometers”

Comment 2: I'm not sure why this model (Wave network) was not in the group of underperforming models, no higher than 76%. Should there just be two groups? With cutoff perhaps at 79%?

Response 2: Thank you for your comment. We agree that it makes sense to only list two groups with a cutoff at 79%. We have revised the manuscript accordingly:

“For clarity, these models are divided into two groups based on their performance. The first group is the group of underperforming models. These models fail to reach an average score higher than 79%. Based on the information in the graph, this group consists of the MLP network ("ONLY DENSE", 75.56%), the small LSTM network ("PURE LSTM", 67.86%), the large LSTM network ("LSTM DENSE", 60.07%), the spatial dropout network ("SPATIAL DROPOUT", 78.93%), and the Wave network ("WAVENET", 71.99%).

The second group consists of three models, performing better than 79% average accuracy: the convolutional network ("CONV1D+DENSE", 79.67%), the dilated convolutional network ("DILATED CNNS", 79.85%) and the LSTM convolutional network ("CONVLSTM2D", 80.37%). These are the good performing networks. For brevity and relevance, only the good performing networks will be considered for the rest of this paper.”

Comment 3: I wonder how accuracy is if combined data from all four sensors used versus just knee? Perhaps can address this in discussion.

Response 3: This is a good suggestion. We list in the future work that we expect the combination of different sensors to make the model more robust, however more work is required on this matter. We have added this to the manuscript:

“Future work will consist of collecting more data and adding more horses to the dataset. This enrichment to the dataset will be used to balance it and allow the model to better discern differences between the activities. It will also balance the impact of excluding Galand from the training data. Another addition is further preprocessing the data, for example by adding filters to eliminate noise and by integrating manually computed features to aid in classification. This could further improve the accuracy of the model. Furthermore data from different sensor locations can be combined. When studying this combination it would be possible to make the model more robust, however more work is required to support this. The findings of this paper are expected to help with the development of wearable horse activity recognition applications and serve as the basis for a framework for horse activity recognition.”

Comment 4: You might note that large intervals would likely work best in dressage or where horse is being ridden in same gait for an extended period of time , versus horses running freely in an enclosure or in the wild where shorter intervals might be better.

Response 4: We agree that it would be useful to mention this. We have revised the manuscript accordingly, the following sentences are added for clarification (lines 350-351):
“Additionally, this logic can be extended to horseback riding as well, where during a dressage session the same gait is exercised for longer periods of time, whereas when horses are in the pasture their movement is less predictable.”

Additionally, more annotations were present in the PDF document, and were all changed according to the suggestions of the reviewer, all changes are highlighted in the revised manuscript.

Reviewer 2 Report

Comments and Suggestions for Authors

Comments and suggestions provided in attached word document.

Author Response

We would like to the reviewer for their insightful, constructive and professional comments.
We have revised the manuscript according to reviewers’ comments.

Please see the attachment for a point-by-point response.
